# Robust Tensor-Based DOA and Polarization Estimation in Conformal Polarization Sensitive Array with Bad Data

**DOI:** 10.3390/s24082485

**Published:** 2024-04-12

**Authors:** Xiaoyu Lan, Lai Jiang, Shuang Ma, Ye Tian, Yupeng Wang, Ershen Wang

**Affiliations:** 1School of Electronic and Information Engineering, Shenyang Aerospace University, Shenyang 110136, China; lanxiaoyu1015@163.com (X.L.); mashuang2@stu.sau.edu.cn (S.M.); tianye171@mails.ucas.ac.cn (Y.T.); ypwang@sau.edu.cn (Y.W.); wes2016@sau.edu.cn (E.W.); 2Key Laboratory of Aerospace Information Sensing and Intelligent Processing Liaoning Province, Shenyang 110136, China; 3State-Owned Changhong Machine Factory, Guilin 541003, China

**Keywords:** conformal polarization sensitive array, direction of arrival and polarization parameters estimation, column vector detection, variational sparse Bayesian learning, tensor

## Abstract

Partially impaired sensor arrays pose a significant challenge in accurately estimating signal parameters. The occurrence of bad data is highly probable, resulting in random loss of source information and substantial performance degradation in parameter estimation. In this paper, a tensor variational sparse Bayesian learning (TVSBL) method is proposed for the estimate of direction of arrival (DOA) and polarization parameters jointly based on a conformal polarization sensitive array (CPSA), taking into account scenarios with the partially impaired sensor array. First, a sparse tensor-based received data model is developed for CPSAs that incorporates bad data. Then, a column vector detection method is proposed to diagnose the positions of the impaired sensors. In scenarios involving partially impaired sensor arrays, a low-rank matrix completion method is employed to recover the random loss of signal information. Finally, variational sparse Bayesian learning (VSBL) and minimum eigenvector methods are utilized sequentially to obtain the DOA and polarization parameters estimation, successively. Furthermore, the Cramér-Rao bound is given for the proposed method. Simulation results validated the effectiveness of the proposed method.

## 1. Introduction

Conformal polarization sensitive arrays (CPSAs) have been found to be attractive in direction of arrival (DOA) and polarization parameters estimation due to their reduced air-friction resistance, reduced array space structure, and strong anti-interference ability [1,2,3,4,5]. However, in real scenarios, huge amounts of sensors are very likely to be affected by harsh natural environments, electromagnetic interference, and component aging, etc., thereby aggravating the impaired probability of the arrays. Thus, the damaged sensors can produce bad data, which means that the effective signal information cannot be received by the impaired sensors. As we all know, traditional subspace methods, such as MUSIC [6,7,8,9] and ESPRIT [10,11], were proposed under the assumption of an intact array structure, and the performance of the above methods highly depends on an accurate correlation matrix structure. If the array output structure information is corrupted, and especially if bad data exist, the DOA estimation performance will dramatically degrade or even fail. Therefore, it is crucial to develop the algorithm for the joint DOA and polarization estimation to reduce the influence of bad data based on a CPSA.

At present, some methods have been proposed in [12,13] to deal with an array with impaired sensors. The basic idea of these methods is that the missing observation data from the impaired sensors can be obtained by reconstructing the array output. A high-dimensional Toeplitz matrix [14] is used to describe the covariance matrix of the uniformly linear array (ULA) with impaired sensors; subsequently, the matrix completion technique is used to recover all the data of the Toeplitz matrix. In this way, the influence of the impaired sensors is eliminated, making it possible to implement DOA estimation using the MUSIC algorithm. Nonetheless, the high complexity of this method wastes a significant amount of time, which originates from searching of the spectral peaks. After detecting the positions of impaired sensors, [15] proposed the method of using a low-rank matrix completion algorithm to recover missing observation data, and then the recursive least square was used to obtain the DOA estimation. Notably, the matrix completion method is only limited to the full-rank matrix and a small parameter estimation error is obtained by sacrificing computational complexity. Unfortunately, it is impractical to replace or repair impaired sensors when large-scale array radar and wireless communication antenna array systems are working. It is known that the performance of DOA estimation is affected by the number of impaired sensors in several ways. As analyzed in [16], an approximate lower bound was proposed as a new metric for the error variance of DOA estimation. Unlike the methods mentioned above, a minimum resource allocation network for the DOA estimation system was proposed in [17], and it can achieve a better estimation performance without knowing the location and time of the impaired sensors when an imperfect array occurs. However, the drawback of this method is that the noise-free and impaired-free training set is difficult to obtain in practice. When faced with complex electromagnetic environments, there is no consideration of the fact that the impaired sensors may also receive a significant number of anomalous signals different than the true signal information, which are distributed in space and are known as bad data. This requires the use of robust algorithms to eliminate the influence of impaired sensors and bad data.

Recently, sparse reconstruction methods [18,19,20,21] have been applied to the theory of DOA estimation and have attracted considerable attention due to their robustness against large background noise. Related methods exploit the sparsity of the incident signals and transform the DOA estimation into a sparse reconstruction problem. Compared with the eigenstructure-based methods, these methods are more adaptable to low signal-to-noise ratios, finite snapshots, spatially adjacent signals, and the correlation of sources. For multiple-input multiple-output (MIMO) radar [22], a robust reweighted $l_{2,1}$-norm penalty method is applied to alleviate the effect of missing observation data due to the impaired sensors, but the disadvantage of this method is that the convergence speed is slow, and it is difficult to find the global optimal solution. In [23], the observations of impaired sensors are considered outliers, and the method of maximum entropy criterion is proposed to mitigate the effect of outliers. Similar to the outliers introduced by impaired sensors, [24] simulated the impulsive noise with outliers and proposed a Bayesian optimization algorithm to achieve robust DOA estimation in terms of accuracy and resolution. It is noted that most of the above research is based on ULA, and it is of considerable interest to extend the idea to CPSAs. Furthermore, the methods mentioned above are all based on matrix form, and the multidimensional structure of the array’s received data is ignored.

In this paper, to enhance the robustness of DOA and polarization parameters estimation performance of CPSAs with bad data, a novel tensor variational sparse Bayesian learning method (TVSBL) is proposed by considering the array received signal model with a three-dimensional (3D) structure. The main contributions of this paper are listed as follows: 

(1) The full electromagnetic vector sensor (EMVS) is used as an array antenna to receive the incident signals. Unlike the conventional scalar sensor, EMVS has the advantage of receiving not only the spatial information of the signal but also the polarization information of the signal, whereas scalar sensors only receive the spatial information of the signal. To fully exploit the multidimensional structural information of the received data and achieve better estimation performance, a tensor-based received signal model is formulated by exploiting the intrinsic 3D features of received data, including spatial, polarization, and temporal fields. In addition, a tensor-based two-dimensional sparse representation of the signal model is given in this paper;

(2) In the case of a partially impaired sensor, the received information is lost randomly with some probability, and bad data will be generated. First, a column vector detection method is employed to diagnose the positions of bad data; furthermore, the low-rank matrix completion method is used to recover the loss of source information. It is worth noting that these strategies can be applied to arbitrary array geometries;

(3) The DOA estimation is performed by the tensor variational sparse Bayesian learning (TVSBL) method with a three-layer hierarchical prior model. After obtaining the estimation value of the DOA, the minimum eigenvector (MME) method is exploited to obtain the polarized angle of incident sources. Meanwhile, the Cramér–RAO bounds are derived for parameter estimation of CPSAs with partially impaired sensors.

The rest of the paper is organized as follows. A tensor-based sparse signal model with bad data is presented in Section 2. Section 3 presents the proposed method for the joint DOA and polarization parameter estimation. Section 4 explores the Cramér–RAO bound for parameter estimation with partially impaired sensors. Section 5 provides the computational load of the proposed method. Section 6 shows the simulation results. Finally, the conclusions are presented in Section 7.

*Notations*: Capital (lowercase) bold-italic notations are used to denote a matrix (vector). ⊗, ⊙, •, and ∘ are the Kronecker product, Khatri–Rao product, Hadamard product and outer product, respectively. (·)^T^, (·)^H^ and (·)^−1^ denote the statistical expectation, transpose, conjugate transpose, and inverse operation, respectively. $\langle \cdot \rangle$ is the statistical operation. ${\left\|\cdot \right\|}_{2}$ is the 2-norm and ${\left\|\cdot \right\|}_{F}$ is the *F*-norm. tr(·) denotes the matrix trace operation. diag (·) denotes a diagonalization matrix with vector elements.

## 2. Problem Statement

### 2.1. Signal Model of CPSA

Assume that there is an arbitrary CPSA with *M* electromagnetic vector sensors (EMVSs) mounted on its surface, as shown in Figure 1. Suppose *K* uncorrelated narrowband far-field sources from azimuth θ and elevation φ are impinging on this CPSA. The spatial-polarization parameter (θk,φk,γk,ηk),k=1,2,⋯K of the incident sources concerning the polarization auxiliary angle γk∈[0,π/2] and the polarization phase difference ηk∈−π,π is defined. In general, the data received by the sensors are analyzed in the global coordinate system. However, because of the curvature of the CPSA, the directional pattern of each sensor has a different orientation. Therefore, it is necessary to transform the orientation pattern matrix from the local coordinate system O′(X′,Y′,Z′) to the global coordinate system O(X,Y,Z) by the Euler rotation transformation matrix R (see [25] for more details). Hence, the array steering vector a^θk,φk,γk,ηk∈ℂ6M×1 [5] of the *k*-th signal is defined as:(1)a^θk,φk,γk,ηk=(g(θk,φk)·a(θk,φk))⊙bθk,φk,γk,ηk=a¯(θk,φk)⊙bθk,φk,γk,ηk
where a¯(θk,φk)=g(θk,φk)·a(θk,φk), g(θk,φk)=[g1(θk,φk),g2(θk,φk),⋯,gM(θk,φk)]T is the *M*×1 sensor pattern in the global coordinate system, and gi(θk,φk) is the response of the *i*-th sensor to the *k*-th signal. a(θk,φk)=[a1(θk,φk),a2(θk,φk),⋯,aM(θk,φk)]T is the *M* × 1 spatial steering vector for the *k*-th signal, and a(θk,φk)=[e−j2π(r1·uk)/λ,⋯,e−j2π(rM·uk)/λ], where λ is the wavelength of the sources, rm=[xm,ym,zm]T,m=1,2,⋯,M and uk=[sinθkcosφk,sinθksinφk,cosθk]T are the sensor position vector and the propagation direction vector of incident sources, respectively. bθk,φk,γk,ηk is the spatial-polarization steering vector [5] of each single EMVS and is given as follows:(2)bθk,φk,γk,ηk=EkHk=exkeykezkhxkhykhzk=cosφkcosθk−sinφksinφkcosφkcosφk−sinθk0−sinφk−cosφkcosθkcosφk−sinφkcosθk0sinθksinγkejηkcosγk
where E=[ex,ey,ez]T and H=[hx,hy,hz]T are the three electric-field components and three magnetic-field components of each complete EMVS [26,27], respectively. Hence, the array received data Xt∈ℂ6M×1 of the *k*-th signal at time *t* is formulated as follows: (3)Xt=∑k=1Ka^θk,φk,γk,ηkskt+nt=A^St+N(t)
where skt is the complex envelope of the *k*-th incoming signal, nt is a white Gaussian noise of zero mean and covariance of σn2. By collecting *T* snapshots, the received data matrix X∈ℂ6M×T is expressed as:(4)X=A^S+N=(A¯⊙B)S+N
where A^=A¯⊙B. A¯=a¯θ1,φ1,a¯θ2,φ2,⋯,a¯θK,φK∈M×K is the spatial steering matrix, B=bθ1,φ1,γ1,η1,⋯,bθK,φK,γK,ηK∈6×K is the spatial-polarization steering matrix of each single EMVS, S=s1(t),s2(t),⋯,sK(t)T∈K×T is the signal matrix, and N(t)=n1(t),n2(t),⋯,n6M(t)T∈6M×T is the Gaussian white noise matrix.

### 2.2. Signal Model with Bad Data

In practice, the sensors usually do not perform as expected, especially when certain occasions, like complex electromagnetic interference, sensor hardware damage, etc. are considered. The potential bad data [28] will be produced by the impaired sensors, which usually have a few large values. Here, those bad data are modeled as additive perturbations on the received array matrix. Thus, the received signal model with bad data can be presented as:(5)$$X=\hat{A}S+N+W$$
where W denotes the bad data produced by the partially impaired sensors, and W(m,:),m=1,2,⋯,M denotes the position of the *m*-th sensor in W. In this paper, the structure of ***W*** is mainly considered from the following scenario: 

(1) When the *m*-th sensor works normally, the values in W(m,:) are all zeros;

(2) When the *m*-th sensor is partially impaired, that means the sensor works randomly and causes random loss of signal information, some of the values in W(m,:) will be non-zeros. The partially impaired sensor receives bad data with a certain probability, and these bad data will be randomly distributed in the W(m,:).

It is seen from (5) that the bad data ***W*** will inevitably cause a great performance degradation or even failure of the parameter estimation algorithm.

## 3. Proposed Method

### 3.1. Tensor-Based Sparse Signal Model with Bad Data

By observing (4), it can be seen that the received array data are stacked in the matrix form, which ignores the inherent 3D structure of the received data. When received data collected from 2D or 3D arrays are inherently organized in a three-dimensional structure, it would be more natural to formulate the data with tensors. Therefore, as the A¯, B and S in (4) have the dimension of M×K, 6×K and K×T, respectively, the tensor-based received signal model with bad data can be rewritten as
(6)X=∑k=1Ka¯(θk,φk)∘b(θk,φk,γk,ηk)∘sk+N+W
where X∈ℂM×6×T. N∈ℂM×6×T represents the noise in the tensor model, which is generated by the tensorization of the traditional Gaussian noise matrix N. W∈ℂM×6×T denotes the bad data produced by the impaired sensors, which is expressed in the tensor model, and W(m,:,:) m=1,2,⋯,M denotes the slice position of the *m*-th sensor in W. 

To better apply the sparse reconstruction class algorithm, a DOA set {(θ˜,φ˜)}={(θ˜1,φ˜1),⋯,(θ˜Jθ˜,φ˜Jφ˜)} with J=Jθ˜Jφ˜≫K  can be obtained by uniformly sampling the ranges of azimuth and elevation [29]. For simplicity, the sparsity of polarization is not considered. Assuming that the true sources are in the DOA set, the tensor-based sparse received signal model with bad data can be rewritten as
(7)X˜=∑k=1K∑j=1Ja˜(θj,φj)∘b˜(θj,φj,γk,ηk)∘s˜j+N˜+W˜
where a˜(θj,φj)=[a1(θj,φj),⋯aM(θj,φj)]T, b˜θ,φ,γk,ηk=[b˜θ1,φ1,γk,ηk,b˜θ2,φ2,γk,ηk,⋯,b˜θJ,φJ,γk,ηk], S˜=s˜1,s˜2,⋯,s˜JT is a J×T complex sparse matrix and only contains *K* non-zero rows. The elements in N∈ℂM×6×T follow additive noise with
(8)p(N)=∏m=1M∏p=16∏t=1TCN(Nmpt0,σ2)
where σ2 is the noise variance, and Nmpt is the (m,p,t) element of N. Bad data W˜mpt are defined as a circularly symmetric complex Gaussian distribution:(9)p(W˜mpt)=∏m=1M∏p=16∏t=1TCN(W˜mpt0,ς)
where ς is the variance of bad data. Then, the Gaussian likelihood function of X˜ has distribution
(10)p(X˜S˜,σ2)=∏m=1M∏p=16∏t=1TCN(X˜mpt∑k=1K∑j=1Ja˜(θj,φj)∘b˜(θj,φj,γk,ηk)∘s˜j,σ2+ς2)
where X˜mpt is the (m,p,t) element of X˜.

### 3.2. Detection of Bad Data Positions

To mitigate the impact of bad data, it is necessary to detect the position of the bad data. In this section, a column vector detection (CVD) method [15,22] is proposed to realize the bad data position detection. Figure 2 shows the slice for each sensor receiving bad data with a random certain probability.

As presented in Figure 2, the received data of the *t*-th snapshot in $\tilde{X}$ is the M×6 matrix $V=\tilde{X}(:,:,t)$. Assume that the *m*-th sensor fails in the *t*-th snapshot, no useful observation data can be obtained except for internal system noise. Meanwhile, bad data will likely be received due to hardware imperfections. Therefore, random noise and bad data are included in the *m*-th row entries of the matrix V. In general, the absolute value of the bad data in the column is much larger than the average of the summed absolute values of the entries in the same column. As a result, the proposed CVD method allows for each column of the matrix V to be traversed and the positions of bad data to be marked as:(11)$$v_{p}(m)>\xi V_{p}$$
where $v_{p}(m)$ denotes the absolute value of the *m*-th entry in the *p*-th column of V; ξ is the weight value, the choice of which is dependent on bad data; $V_{p}$ is the mean of the absolute values of all *M* entries in the *p*-th column and can be calculated by
(12)$$V_{p}=\frac{1}{M}\sum_{m=1}^{M}v_{p}(m)$$

As can be seen from (11), some entries in V are determined to be bad data if their value is above the selected average threshold; then, the positions that present bad data are replaced by zero [20]
(13)$$\Gamma_{\Omega }\left[V(m,p)\right]=\left\{\begin{array}{c} V(m,p)\\ 0 \end{array}\right.\begin{array}{c} (m,p)\notin \Xi \\ otherwise \end{array}$$
where $\Gamma_{\Omega }\left[V(m,p)\right]$ represents the observation operator for V(m,p) and Ξ is the position subset containing bad data. The proposed CVD method is outlined as Algorithm 1 below.
**Algorithm 1: Column Vector Detection Method (CVD)**Step 1. Calculated $V_{p}$ by (12)Step 2. **Initialization:** the weight value ξ=2 and the position subset $\Xi^{0}=\phi$, where ϕ is a null index setStep 3. **For** each diploes *p* (*p* = 1, 2, …, 6)Step 4.    **For** each sensor *m* (*m* = 1, 2, …, *M*)Step 5.      Find the error data positions by (11)Step 6.      Replace the error data positions of V(m,p) with zeroStep 7.    **End**Step 8.    Update the index set by finding the zero position: $\Xi^{p}=m$Step 9. **End**Step 10. **Output:** the position subset Ξ

### 3.3. Recovery of Lost Signal Information

After finding the position of the bad data in $\tilde{X}$ by the CVD method, the new tensor-based received array data $\overset{⌣}{X}$ are obtained by filling the position of the bad data with zero. To simplify the calculation, expanding $\overset{⌣}{X}$ in matrix form yields $\tilde{X}={ℑ}^{\left(1\right)}[\overset{⌣}{X}]$, respectively, where ${\left\{{ℑ}^{\left(i\right)}[•]\right\}}_{i=1,2,3}$ is the third-order tensor unfolding along the *i*-th mode [30]. For a partially impaired sensor array, it is obvious that $\tilde{X}$ is a full-rank matrix because the inconsecutive zero elements are randomly distributed. Poor accuracy in the estimation of parameters will occur if there are a lot of zero elements in $\tilde{X}$. Therefore, the lost signal recovery is necessary to improve robustness against bad data. 

The incomplete received matrix X˜ can be regarded as a problem of low-rank matrix completion (LRMC) [31]. A complete and denoised matrix X^ with rank r<min{6M,T} (r=K) is expected to be obtained from the incomplete noise full-rank l matrix X˜ (l=min{6M,T}). In this subsection, a signal recovery method is proposed to construct a complete low-rank received signal matrix X^ by minimizing the Frobenius norm [32]. Consider that the low-rank r matrix X˜ can be decomposed into X˜=FG, where F∈ℂ6M×r and G∈ℂr×T are random matrices. Thereafter, the matrix completion achieved by non-convex relaxation can be presented as:(14)minF,G,H12H−FGF2 , subject to ΓΞ[H]=ΓΞ[X˜]
where H∈ℂ6M×T. Under the premise of minimizing (14), each variable F, G, and H is updated by fixing the other two. Thus, the iterative procedure of the LRMC method is summarized as follows:(15)Fi+1=HiGi+=argminF12Hi−FGiF2
(16)Gi+1=Fi+1+Hi=argminG12Hi−Fi+1GiF2
(17)Hi+1=Fi+1Gi+1+ΓΞ(X˜−Fi+1Gi+1)
where Fi, Gi, and Hi denote the output results of matrices F, G, and H at the *i*-th iteration. This method will stop the iteration until the following criteria have been satisfied:(18)Hi+1−Fi+1Gi+1F2Hi+1F2≤δ
where the δ is set to 10−6. In the end, a complete and denoise matrix low-rank matrix X^ is obtained by:(19)X^=Fi+1Gi+1

The main steps of the low-rank matrix completion method are described in Algorithm 2 below.
**Algorithm 2: Low Rank Matrix Completion Method (LRMC)**Step 1. **Initialize:** Generate the random matrix F0∈ℂ6M×r and G0∈ℂr×T, and H0=ΓΞ(X˜)∈ℂ6M×TStep 2. **While** not converged do      Update Fi+1, Gi+1 and Hi+1 by (15)~(17)     i=i+1Step 3. End reaches the stopping criteria in (18)Step 4. **Output:** A complete and denoise matrix low-rank matrix X^=Fi+1Gi+1

### 3.4. The DOA Estimation Method via TVBSL

After the position detection of the bad data and the recovery of the lost information, the complete and denoise array output matrix is X^, and the next step is to estimate the DOA accurately. More recently, variational sparse Bayesian learning (VSBL) has been explored to give deterministic approximations of posteriors with high accuracy and low complexity, especially in the case of data perturbation. Therefore, to further confront the effect of the bad data and fully utilize the multidimensional information of the array received data, the VSBL technique based on the tensor model is proposed to improve the performance of DOA estimation. First, let us reshape the estimation X^ into a tensor of X^∈ℂM×6×T, the posterior distribution of S˜ can be calculated by the Bayesian criterion [33], which is expressed as:(20)pS˜,α,βX^=pX^,S˜,α,βp(X^)

To improve the performance of the Bayesian algorithm, a novel three-layer hierarchical prior model is formulated to promote sparse solutions [33,34]. First, assume that the elements of the first layer prior $p(\tilde{S}\left|\alpha \right.)$ follow a zero-mean complex Gaussian distribution:(21)$$p(\tilde{S}\left|\alpha \right.)=\prod_{t=1}^{T}CN(\tilde{S}(t){\left|0,\Lambda \right.}^{-1})=\prod_{t=1}^{T}\prod_{j=1}^{J}CN({\tilde{S}}_{j}(t)\left|0,\alpha_{j}\right.)$$
where $\Lambda^{-1}=diag(\alpha )=diag(\alpha_{1},\alpha_{2},\cdots ,\alpha_{J})$ is the noise precision for different rows in $\tilde{S}$. Second, each hyperparameter $\alpha_{j}$ in α follows an exponential (Exp) distribution with the parameter as the second layer prior:(22)$$p(\alpha \left|\tau ,\beta \right.)=\prod_{j=1}^{J}Exp(\alpha_{j}\left|\tau \beta \right.)$$

Similarly, the parameter β is modeled as the chi-square (Chi2) distribution of the third layer prior:(23)$$p(\beta \left|v\right.)=\chi^{2}(\beta \left|v\right.)$$

In the Bayesian framework, the true posterior distribution is unsolvable due to the need for the integral operation. To handle this issue, the VSBL method is exploited in this paper to provide a deterministic approximation of posteriors with high accuracy. Let Θ be a set containing the latent variable $\tilde{S}$ and hyperparameters (α and β) in the probabilistic model. The distribution q(Θ) can be closely approximated as the true posterior distribution $p(\Theta \left|\hat{X}\right.)$ by minimizing the Kullback–Leibler distance [35]:(24)$$p(\tilde{S},\alpha ,\beta \left|\hat{X}\right.)\approx q(\tilde{S},\alpha ,\beta )=q(\tilde{S})q(\alpha )q(\beta )$$

By removing the term α from Θ, $q(\tilde{S})$ can be calculated by $lnq(\tilde{S})\propto ln{\langle p\left(\hat{X},\tilde{S},\alpha ,\beta \right)\rangle }_{q(\alpha )}$, and the approximate posterior of $q(\tilde{S})$ can be obtained as:(25)$$\begin{array}{l} \ln q\left(\tilde{S}\right)∼{\langle \sum_{t=1}^{T}-\frac{1}{2\sigma^{2}}{\left\|\tilde{x}\left(t\right)-\tilde{A}s\left(t\right)\right\|}_{2}^{2}+\sum_{t=1}^{K}\sum_{j=1}^{J}\frac{s_{j}^{2}\left(t\right)}{2\alpha_{j}}\rangle }_{q\left(\alpha \right)}\\ \Rightarrow q\left(\tilde{S}\right)∼\prod_{t=1}^{T}\exp\left(-\frac{1}{2}{\tilde{S}}^{H}(t)\Gamma^{-1}\tilde{S}(t)+{\tilde{S}}^{H}(t)\Gamma^{-1}\mu (t)\right)\\ \;\;\;∼\prod_{t=1}^{T}CN({\tilde{S}}^{H}\left(t\right)\left|\mu (t),\Gamma \right.) \end{array}$$
where the parameters μ(t) and Γ can be calculated according to:(26)$$\mu (t)=\sigma^{-2}\Gamma^{-1}{\bar{A}}^{H}({ℑ}^{\left(1\right)}\left[\hat{X}\right](t))$$
(27)$$\Gamma ={\left(\sigma^{-2}{\tilde{A}}^{H}\tilde{A}+\langle \Lambda \rangle \right)}^{-1}$$

The proof process of (26) and (27) is shown in Appendix A. q(α) can be written as the product of a generalized inverse Gaussian distribution and can be updated by $q(\alpha )\propto {\langle p(\tilde{S}\left|\alpha \right.)p(\alpha \left|\tau ,\beta \right.)\rangle }_{q(\tilde{S})q(\beta )}$;
(28)$$\begin{array}{l} \ln q(\alpha )∼{\langle \ln\left[\prod_{t=1}^{K}\prod_{j=1}^{J}\frac{1}{\sqrt{2\pi \alpha_{j}}}\exp\left(-\frac{h_{j}^{2}\left(t\right)}{2\alpha_{j}}\right)\prod_{j=1}^{J}\exp(-\tau \beta \alpha_{j})\right]\rangle }_{q(\tilde{S})q(\beta )}\\ \Rightarrow q(\alpha )∼\prod_{j=1}^{J}\alpha_{j}^{-\frac{1}{2}}\exp\left(-\frac{h_{j}^{2}\left(t\right)}{2\alpha_{j}}-\tau <\beta >\alpha_{j}\right) \end{array}$$

Thus, we have
(29)$$\langle \alpha_{j}^{n}\rangle ={\left(\frac{\langle h_{j}^{2}\rangle }{2\tau \langle \beta \rangle }\right)}^{\frac{n}{2}}\frac{\kappa_{n-\frac{1}{2}}\left(\sqrt{2\tau \langle \beta \rangle \langle h_{j}^{2}\rangle }\right)}{\kappa_{-\frac{1}{2}}\left(\sqrt{2\tau \langle \beta \rangle \langle h_{j}^{2}\rangle }\right)}$$
where $\langle h_{j}^{2}\rangle =\mu_{j}^{2}+\Gamma_{j}$; and $\kappa_{p}(\cdot )$ is a Bessel function of the third kind with order *p*. Setting n=−1 gives the estimated Γ and $\langle \Lambda \rangle$. The posterior distribution q(β) can be calculated by $q(\beta )\propto {\langle p(\alpha \left|\tau ,\beta \right.)p(\beta \left|v\right.)\rangle }_{q(\alpha )}$
(30)$$\begin{array}{l} \ln q(\beta )∼{\langle \ln\left[\prod_{j=1}^{J}\tau \beta \exp(-\tau \beta \alpha_{j})\frac{2^{-\frac{v}{2}}}{\Delta (\frac{v}{2})}\beta^{-\frac{v}{2}-1}\exp(-\frac{\beta }{2})\right]\rangle }_{q\left(\alpha \right)}\\ \Rightarrow q(\beta )∼\beta^{J+\frac{v}{2}-1}\exp\left(-\tau \sum_{j=1}^{J}<\alpha_{j}>\beta -\frac{\beta }{2}\right) \end{array}$$

Thus, β follows the gamma distribution.
(31)$$\beta \sim Gamma\left(J+\frac{\nu }{2},\tau \sum_{j=1}^{J}\langle \alpha_{j}\rangle +\frac{1}{2}\right)$$

Therefore, the mean value of β can be obtained as:(32)$$\langle \beta \rangle =\frac{J+\frac{v}{2}}{\tau \sum_{j=1}^{J}\langle \alpha_{j}\rangle +\frac{1}{2}}$$

The proposed TVSBL method outputs the mean and variance of the recovery signal until the iteration converges. The power spectrum can be represented as:(33)$$\hat{P}=\frac{1}{L}\langle {\left\|\tilde{S}\right\|}_{2}^{2}\rangle =\frac{1}{L}\left({\left\|\langle \tilde{S}\rangle \right\|}_{2}^{2}+\langle {\left\|\tilde{S}-\langle \tilde{S}\rangle \right\|}_{2}^{2}\rangle \right)=\frac{{\left\|\mu \right\|}_{2}^{2}}{L}+\Gamma$$
where the *K* largest peak positions of the power spectrum represent the estimated DOAs $\left\{({\hat{\theta }}_{1},{\hat{\varphi }}_{1}),({\hat{\theta }}_{2},{\hat{\varphi }}_{2}),\cdots ,({\hat{\theta }}_{K},{\hat{\varphi }}_{K})\right\}$. The procedure for the proposed TVSBL method is summarized in Algorithm 3.
**Algorithm 3: Tensor Variational Sparse Bayesian Learning (TVSBL) Method**Step 1. **Input:** the steering matrix $\tilde{A}$, observation data matrix $\hat{X}$Step 2. **Initialize:** the hyperparameters $\langle \alpha_{j}\rangle$, $\langle \alpha_{j}^{-1}\rangle$ and $\langle \beta \rangle$, the rate parameter τ, the shape parameter v, the variance $\sigma^{2}$, the threshold ε, the number of iterations $i_{\max}$Step 3. **while** not converged doStep 4.  Update variance $\Gamma^{\left(i\right)}$ by (27)Step 5.  Update mean $\mu {\left(t\right)}^{\left(i\right)}$ by (26)Step 6.  Update hyperparameter ${\langle \alpha_{j}{\;}^{-1}\rangle }^{\left(i\right)}$ by (29)Step 7.  Update hyperparameter ${\langle \beta \rangle }^{\left(i\right)}$ by (32)Step 8. **End while**Step 9. **Output**: The mean μ and variance Γ of the reconstruction signal

**Remark** **1.** In [36,37], Gaussian-Gamma and Laplace are chosen as two layers of the posterior distribution. The Gaussian-Exp-Chi2 (GEC) distribution proposed in this paper exhibits a sharp peak at the origin and heavy tails. Thus, the GEC distribution enforces the sparsity of the solution to a greater extent than other conventional distributions.

### 3.5. MME-Based Polarization Parameter Estimation Method

To facilitate the MME to obtain the polarization parameters, the matrix-received data model is constructed by utilizing (1) (2) and (3)
(34)$$X=\sum_{k=1}^{K}\left[\bar{a}(\theta_{k},\varphi_{k})⊙b(\theta_{k},\varphi_{k},\gamma_{k},\eta_{k})\right]s_{k}$$

By defining the array output covariance matrix $Q=\frac{1}{T}XX^{H}$, the signal subspace can be obtained by the eigenvalue decomposition of Q [38,39]. Hence, the noise subspace can be expressed as $U_{N}=I_{6M}-U_{S}U_{S}^{H}$. Next, substituting the *K* estimated DOAs into the constructed spatial spectrum function, *K* functions can be obtained, as follows:(35)$$W={\bar{A}}^{H}\left(\theta ,\varphi ,\gamma ,\eta \right)U_{N}\bar{A}\left(\theta ,\varphi ,\gamma ,\eta \right)$$

The polarization parameters can be calculated by the eigenvector corresponding to the minimum eigenvalue of W
(36)$$\gamma_{k}=\arctan\left(\mathrm{abs}\left(\frac{\rho_{k}\left(2\right)}{\rho_{k}\left(1\right)}\right)\right)$$
(37)$$\eta_{k}=\mathrm{angle}\left(\frac{\rho_{k}\left(2\right)}{\rho_{k}\left(1\right)}\right)$$
where $\rho_{k}\left(i\right)$ is the *i*-th element in the *k*-th eigenvector.

In this paper, a column vector detection algorithm, low-rank matrix completion algorithm, tensor variational sparse Bayesian learning algorithm and minimum eigenvector algorithm are successively used. To facilitate an understanding of the execution steps of the proposed method, a flowchart is shown in Figure 3.

## 4. Cramér–Rao Bound

For the observation model and the proposed method, CRB can be regarded as a lower bound of the incident signal DOA estimation [1]. It can be calculated by taking the inverse of the Fisher information matrix (FIM). For the 2D DOA estimation for the CPSA with sensor failure, the incident signal can be described by the set h, including all the unknown parameters
(38)$$h=[\theta_{1},\theta_{2},\cdots ,\theta_{K},\varphi_{1},\varphi_{2},\cdots ,\varphi_{K}]$$

Thus, the FIM can be specifically formulated as:(39)$$F=\left[\begin{array}{cc} F_{\theta \theta } & F_{\theta \varphi }\\ F_{\varphi \theta } & F_{\varphi \varphi } \end{array}\right]$$

In general, the distribution of the incident signal matrix received data model can be written as:(40)$$Y∼CN(0,R_{Y})$$
where $Y={ℑ}^{\left(i\right)}[\hat{X}]$, $R_{Y}=\Upsilon AR_{S}A^{H}\Upsilon^{H}+\sigma^{2}I_{6M}$, and $R_{S}=SS^{H}$. Υ is the *M × N* matrix, and its (*m*, *n*)-th position on the main diagonal takes zero if the *n*-th channel of the *m*-th sensor fails. Thus, the log-likelihood function of every concerning column is:(41)$$L(h)=L*\{ln(R_{Y})+tr(R_{Y}{\;}^{-1}R_{Y})\}$$

The element of FIM can be obtained by calculating the second derivative of L(h) with respect to θ and φ
(42)$$F_{\theta \varphi }=L*tr\{R_{Y}{\;}^{-1}\frac{\partial R_{Y}}{\partial h_{\theta }}R_{Y}{\;}^{-1}\frac{\partial R_{Y}}{\partial h_{\varphi }}\}$$
where the first-order derivative of $R_{Y}$ with respect to $h_{\theta }$ can be computed as:(43)$$\frac{\partial R_{Y}}{\partial h_{\theta }}=B_{\theta }R_{S}{\left(\Upsilon A\right)}^{H}+\Upsilon AR_{S}B_{\theta }^{H}$$
where $B_{\theta }=\frac{\partial (\Upsilon A)}{\partial h_{\theta }}=\frac{\partial (\Upsilon (A_{xoy}⊙A_{z}))}{\partial h_{\theta }}$. $A=A_{xoy}⊙A_{z}$, $A_{xoy}$ and $A_{z}$ is the steering vector of the xoy-planar and the steering vector along the z-axis, respectively. After substituting (43) into (42), the $F_{\theta \varphi }$ can be obtained by
(44)Fθφ=2L∗Re{tr[RY−1BθRS(ΥA)HRY−1ΥARSBφH]+tr[RY−1BθRS(ΥA)HRS−1BφRS(ΥA)H]}

The result $B_{\theta }$ of the derivative of ΥA about θ can be computed as
(45)$$B_{\theta }=\frac{\partial (\Upsilon (A_{xoy}⊙A_{z}))}{\partial \theta_{m}}=\Upsilon (A_{z}⊙{\overset{\cdot }{A}}_{xoy})u_{\theta_{m}}u_{\theta_{m}}{\;}^{T},\;m=1,2,\cdots ,K$$
where
(46)$${\overset{\cdot }{A}}_{xoy}=\frac{\partial A_{xoy}}{\partial \theta_{m}}[a_{xoy}(\theta_{1})•d_{xoy}(\theta_{1}),\cdots ,a_{xoy}(\theta_{K})•d_{xoy}(\theta_{K})]$$
(47)$$d_{xoy}(\theta_{k})=j\frac{2\pi }{\lambda }[(r_{x_{1}}\sin\varphi_{k}\sin\varphi_{k}+r_{y_{1}}\sin\varphi_{k}\cos\theta_{k}),\cdots ,(r_{x_{M_{c}}}\sin\varphi_{k}\sin\varphi_{k}+r_{y_{M_{c}}}\sin\varphi_{k}\cos\theta_{k})]$$
and $u_{\theta_{m}}$ represents the $\theta_{m}$-th column of the identity matrix $I_{K}$. Let $U_{xoy}=\Upsilon (A_{z}⊙{\overset{\cdot }{A}}_{xoy})$, and then the submatrix $F_{\theta \theta }$ in F can be rewritten as the matrix form, thus
(48)Fθθ=2L∗Re{[RS(ΥA)HRY−1ΥARS]•[U xoyHRY−1Uxoy]+[RS(ΥA)HRY−1Uxoy]•[RS(ΥA)HRY−1Uxoy]}
where $F_{\theta \theta }$ is the DOA estimation corresponding to the xoy-planar. In the same way, the submatrix in F corresponding to the z-axis and the cross-submatrix can be obtained by calculating the following equations:(49)Fφφ=2L∗Re{[RS(ΥA)HRY−1ΥARS]•[U zHRY−1Uz]T+[RS(ΥA)HRY−1Uz]•[RS(ΥA)HRY−1Uz]T}
(50)Fθφ=2L∗Re{[RS(ΥA)HRY−1ΥARS]•[UzHRY−1Uxoy]T+[RS(ΥA)HRY−1Uz]•[RS(ΥA)HRY−1Uxoy]T}
(51)Fφθ=2L∗Re{[RS(ΥA)HRY−1ΥARS]•[U xoyHRY−1Uz]T+[RS(ΥA)HRY−1Uxoy]•[RS(ΥA)HRY−1Uz]T}
where
(52)$$U_{z}=\Upsilon ({\overset{\cdot }{A}}_{z}⊙A_{xoy})$$
(53)$${\overset{\cdot }{A}}_{z}=\frac{\partial A_{z}}{\partial \varphi_{m}}=[a_{z}(\varphi_{1})•d_{z}(\varphi_{1}),\cdots ,a_{z}(\varphi_{K})•d_{z}(\varphi_{K})]\;m=1,2,\cdots K$$
(54)$$d_{z}(\varphi_{k})=j\frac{2\pi }{\lambda }[r_{z_{1}}\sin\varphi_{k},\cdots ,r_{z_{M_{z}}}\sin\varphi_{k}]$$

As a result, based on the above (48)–(51), the CRB of the DOA estimation for partially impaired sensors existing in the array can be presented as:(55)$$CRB=F^{-1}={\left(F_{\theta \theta }-F_{\theta \varphi }F_{\varphi \varphi }^{-1}F_{\varphi \theta }^{T}\right)}^{-1}$$

## 5. Complexity Analysis

For each iteration of the latent variables and hyperparameters stated above, it is easy to see that the computational complexity of the proposed method mainly derives from the following steps: (1) constructing the tensor-received data model; (2) updating the mean μ for $\tilde{S}$; and (3) updating the variance Γ for $\tilde{S}$. The specific complexity can be expressed as follows: (1) $6MLJ^{2}$, (2) $2J^{3}-2J^{2}+36MK^{2}J^{2}$, (3) $J^{3}+12MK^{2}J^{2}$. Therefore, the total complexity of the proposed method is $3J^{3}-2J^{2}+48MK^{2}J^{2}+6MLJ^{2}$.

As a comparison, the variational sparse Bayesian learning algorithm proposed in [5] is abbreviated as VSBL, which requires $3J^{3}-2J^{2}+48MK^{2}J^{2}$, and the sparse Bayesian learning algorithm proposed in [40] is abbreviated as SBL, which requires $J^{3}+J^{2}T+18MK^{2}J^{2}$. Ref. [37] is an extension algorithm for off-grid angle estimation, abbreviated as OGSBL, which requires $J^{3}+18MK^{2}J^{2}$.

## 6. Simulations

### 6.1. Simulation Settings 

In this section, we evaluate the performance of the proposed method by numerical simulations through estimation accuracy and resolution under the cases of partially impaired sensors and compare the TVSBL with that of VSBL [5], OGSBL [37], and SBL [40]. In the following simulations, the estimation accuracy is evaluated by root mean square error (RMSE), and the resolution is evaluated by the probability of successful detection. The “successful trial” is defined in this paper as $\left|\hat{\theta }-\theta \right|<\Delta \theta /2$, where $\hat{\theta }$ is the estimated value, θ is the true value, and Δθ is the angle interval between $\hat{\theta }$ and θ.

As shown in Figure 4, $M_{c}\times M_{z}$ sensors are uniformly distributed on the surface of the cylindrical CPSA. The spacing of the sensors along the Z-axis is λ/2. For simplicity, assume that $M_{c}=5$ and $M_{z}=5$. The ranges of the azimuth and elevation are sampled with a 1° interval to form the DOA set $\{(\tilde{\theta },\tilde{\varphi })\}$. Two uncorrelated incident signals with spatial parameters (θ,φ) from $\left(-17.2^{\circ },20.2^{\circ }\right)$ and $(10.9^{\circ },60.1^{\circ })$ are considered, and the polarization parameters (γ,η) are located at $(55.3^{\circ },65.7^{\circ })$ and $(35.4^{\circ },25.1^{\circ })$, respectively.

In the resolution experiment, the azimuth of the closed target is selected as $\left(-3.9^{\circ },4.1^{\circ }\right)$. The bad data are drawn from under the following scenario: a partially impaired sensor array received bad data with a probability of Ω=0.05. The variance of bad data ς is set as 100. The proposed TVSBL method is initialized by setting $\alpha =\frac{1}{L}\sum_{l=1}^{L}\tilde{A}{ℑ}^{\left(1\right)}[\hat{X}]$, β=0.1, τ=1.5, and v=1. The RMSE is defined as:(56)$$RMSE=\sqrt{\frac{1}{LK}\sum_{t=1}^{T}\sum_{k=1}^{K}{\left({\hat{u}}_{kt}-u_{k}\right)}^{2}}$$
where *L* denotes the number of Monte Carlo simulations, $u_{k}$ is one of the parameters of $\left(\theta_{k},\varphi_{k},\gamma_{k},\eta_{k}\right)$, and ${\hat{u}}_{kt}$ is an estimate of $u_{k}$ in the *t*-th simulation. *L* is set to 200 in the simulation.

The simulation results were obtained on a PC with MATLAB R2021b, Intel Core i5 @3.0 GHz processor, and 16 GB LPDDR3 @ 6000 MHz.

### 6.2. Simulation Results and Analysis 

Figure 5 and Figure 6 illustrate the DOA and polarization estimation RMSEs of various algorithms versus SNR with snapshots = 20, respectively. Here, assume that every sensor is partially impaired and receives bad data with a probability Ω is 0.05. As shown in Figure 5 and Figure 6, both the SBL and OGSBL algorithms have a similar RMSE performance, whereas they provide the worst performance in the whole SNR regions in the condition where the sensor is partially impaired. This is due to the effect of the existence of bad data; the SBL and OGSBL algorithms are impressionable with respect to the mismatch in the sparse presentation model. Compared with the SBL and OGSBL methods, the VSBL based methods, i.e., VSBL and the proposed TVSBL method, exhibit better RMSE performance, as expected, because the VSBL-like methods are not as susceptible to data perturbations. Moreover, it can be observed that the performance of the proposed method, which benefits from the recovery of bad data, is superior to the conventional VSBL method. Nevertheless, there is still a gap between their RMSEs and the CRB. It is important to point out that, unlike the conventional DOA estimation methods, the proposed method is framed by the integration of bad data detection and recovery and the sparse signal presentation based on the tensor model. This is the key mechanism making the proposed method perform a better DOA and polarization estimation in the presence of a sensor partially impaired for the CPSA.

Figure 7 and Figure 8 illustrate the RMSEs of DOA and polarization versus the number of snapshots for different methods, while the SNR is set to 3 dB and the other parameters remain the same as those in Figure 5 and Figure 6. As shown in Figure 7 and Figure 8, the estimation performance of all four methods improves when increasing the number of snapshots. It is further evident that the proposed method provides a competitive advantage in performing the DOA and polarization estimation in the condition of the sensor being partially impaired by varying the snapshots.

Figure 9 and Figure 10 present the probability of the successful detection (PSD) performance versus SNR, and the number of snapshots, respectively. In Figure 9, the number of snapshots is fixed to 20. In Figure 10, the SNR is set to 3 dB. The other parameters remain the same as before. In this simulation, the definition of one “successful detection” is illustrated in Section 6.1. It is shown that the PSDs of SBL and OGSBL are rather low when the SNR is lower, or the snapshots are smaller. Especially in Figure 10, when the SNR is 3dB, although the snapshots are increasing, the SBL and OGSBL algorithms still fail to distinguish the two emitting signals. Compared with SBL and OGSBL, the VSBL and the proposed method possess higher PSDs, which means that the resolution of the two algorithms is much higher. As expected, the resolution performance of the proposed method is the best; the PSD curve of the proposed method is much higher than that of the VSBL, even with a relatively lower SNR and smaller snapshots. Figure 9 and Figure 10 further verify that our proposed method is robust to bad data in sensors.

Figure 11 and Figure 12 present a comparison of the DOA RMSE of the proposed method versus SNR and snapshots under different Ω, respectively. In Figure 11, the number of snapshots is fixed to 20. In Figure 12, the SNR is fixed to 3 dB. It is obvious that the DOA RMSE performance improves as the SNR and snapshots increase. On the other hand, the RMSE will rise when the Ω is larger. In other words, the higher the probability of bad data, the lower the estimation accuracy.

To fully explore the effect of bad data on the performance of the four algorithms, the RMSE of DOA and polarization estimation with different ς are shown in Figure 13 and Figure 14, respectively. As illustrated in 3.1, the ς denotes the variance of the bad data. The greater the value, the stronger the effect of the bad data on useful information. The snapshots are set to 20 and the SNR is set to 3 dB in Figure 13 and Figure 14. The probability Ω is 0.05. Figure 13 and Figure 14 show that the DOA and polarization RMSE of the four methods deteriorate when the snapshots increase. However, the proposed algorithm exhibits the smoothest curve, and its estimation accuracy significantly surpasses that of the other three methods. This simulation result validates the effectiveness of the proposed method in mitigating the impact of partially impaired sensors on the DOA and polarization estimation performance within the CPSA. Additionally, it demonstrates superior tolerance and robustness against erroneous data.

To further verify the robustness of the proposed method, two cases of array element position error and array element mutual coupling will be considered in the following simulation. Figure 15 illustrates the DOA RMSE versus SNR while the snapshots are 20 in the condition of the array element position error, and the element position error is randomly chosen by Perror∈[−0.4,0.4]λ/2. It can be seen in Figure 15 that the RMSE of the proposed algorithm with the element position error is only slightly lower than that without the element position error. Moreover, the RMSE of the proposed algorithm is much lower than that of the other three algorithms without the element position error. This fully shows that the proposed algorithm can still maintain relatively high estimation accuracy under the influence of the element position error, thus proving that the proposed algorithm is relatively robust. Figure 16 illustrates the DOA RMSE versus SNR, while the snapshots are 20 in the condition of array element mutual coupling. In this simulation, it is assumed that only the linear array placed along the z axis has a mutual coupling effect, and the three linear arrays are far apart due to curvature; therefore, the mutual coupling effect between the linear arrays is ignored. When the mutual coupling matrix is set to C=Toeplitz{[1,0.3527+0.1854j,0,0,0]}∈ℂMz×Mz; it is obvious to see that the RMSE of the proposed method is close to that of the proposed method without considering the mutual coupling. The above experiments demonstrate the robustness of the proposed method. When the SNR is higher, the mutual coupling error has almost no effect on the proposed algorithm, which further proves that the proposed algorithm has strong robustness.

## 7. Conclusions

In this paper, the problem of CPSAs with partially impaired sensor arrays was addressed. For the first time, a tensor-based model of received data for CPSAs, which benefits from spatial, polarization, and temporal diversity of incident sources, was established, and the effectiveness of the model was verified in the presence of bad data received by impaired sensors. To eliminate the effect of bad data, the CVD method was used to detect the locations of bad data and set them to zero. Then, the missing signal information of the array output matrix was processed by removal and low-rank matrix completion methods. Finally, the TVSBL and the MME method were proposed to obtain the DOA and polarization parameters estimation, respectively. In the simulation results, the proposed model and method show many advantages over existing methods in terms of estimation accuracy and resolution in partially impaired sensor array scenarios, while the comparison methods are inferior to the proposed method. The limitation of the proposed method is that, when the SNR is 0dB, there is no obvious difference between the proposed method and the comparison methods in terms of parameter estimation performance. In the future, we will focus on how to optimize the performance of the sparse method at a low SNR.

## Figures and Tables

**Figure 1 sensors-24-02485-f001:**
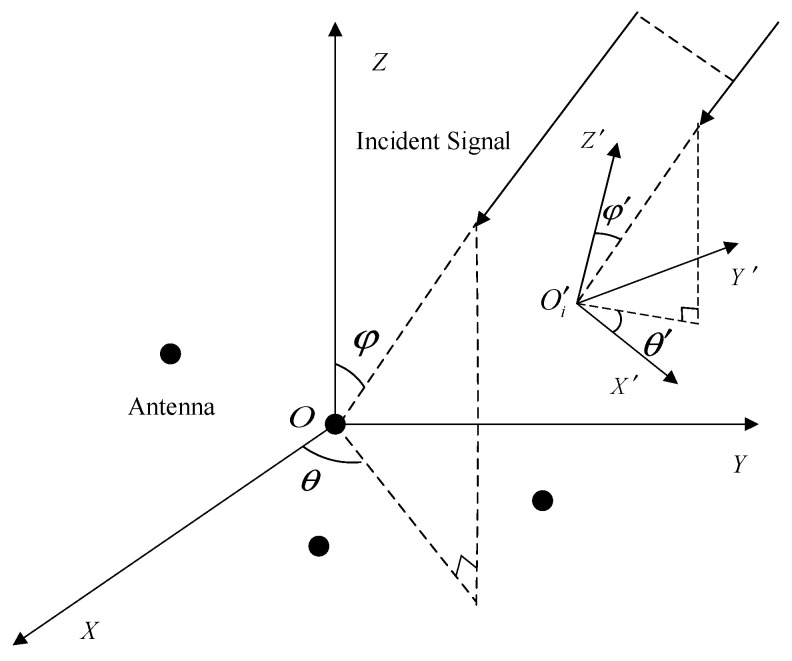
Arbitrary structure of a CPSA.

**Figure 2 sensors-24-02485-f002:**
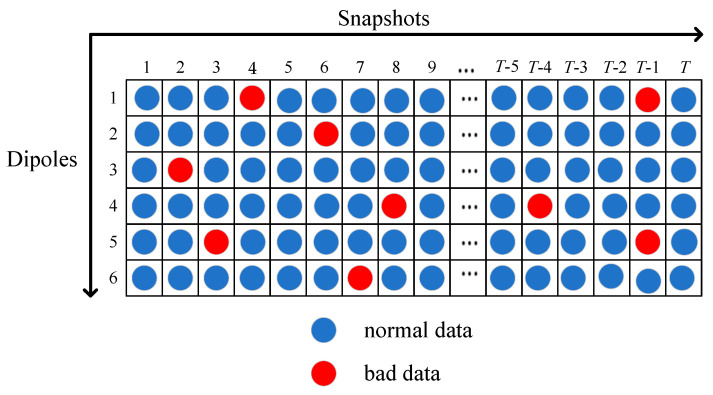
Slice for each sensor receiving bad data with a random certain probability.

**Figure 3 sensors-24-02485-f003:**
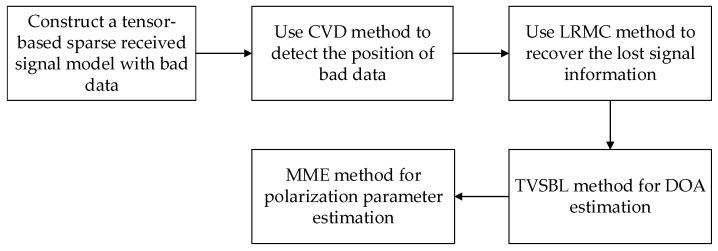
A flowchart of the proposed method.

**Figure 4 sensors-24-02485-f004:**
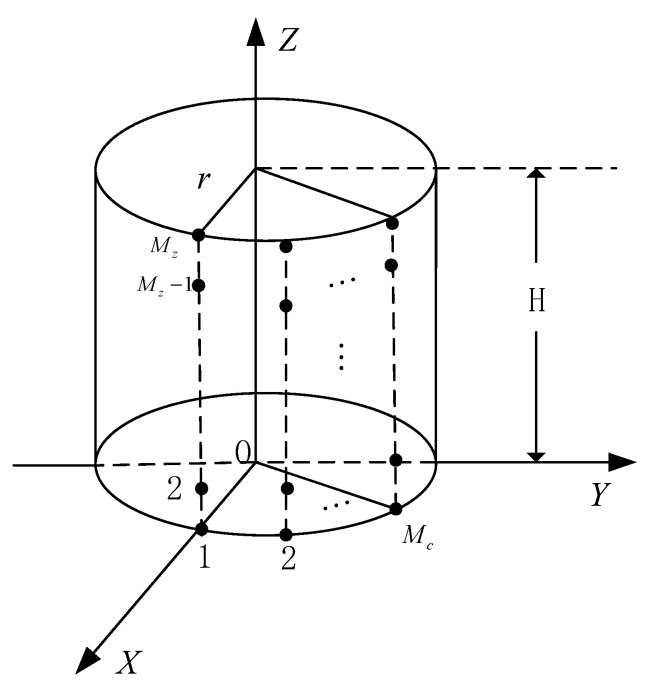
The array structure of the cylindrical CPSA.

**Figure 5 sensors-24-02485-f005:**
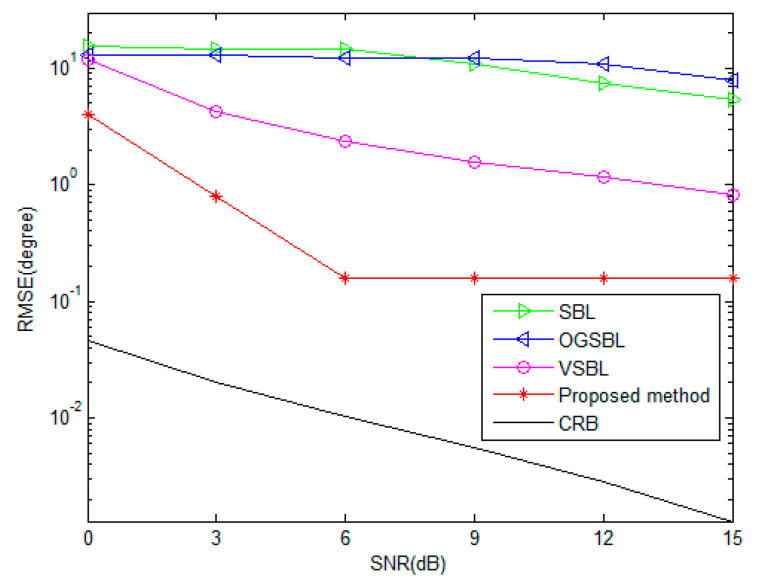
DOA RMSE versus SNR (snapshots = 20).

**Figure 6 sensors-24-02485-f006:**
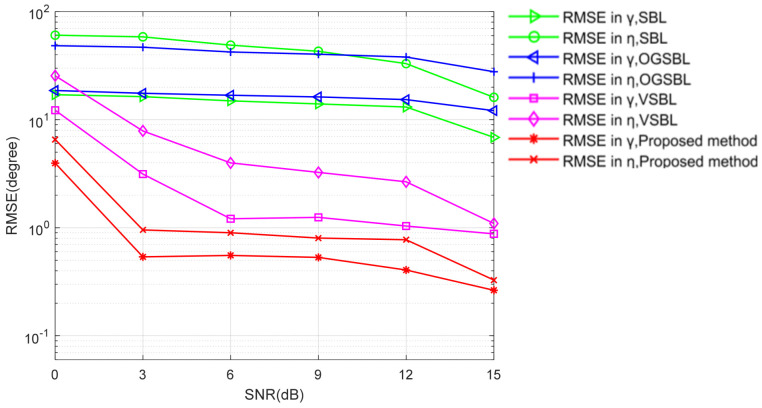
Polarization RMSE versus SNR (snapshots = 20).

**Figure 7 sensors-24-02485-f007:**
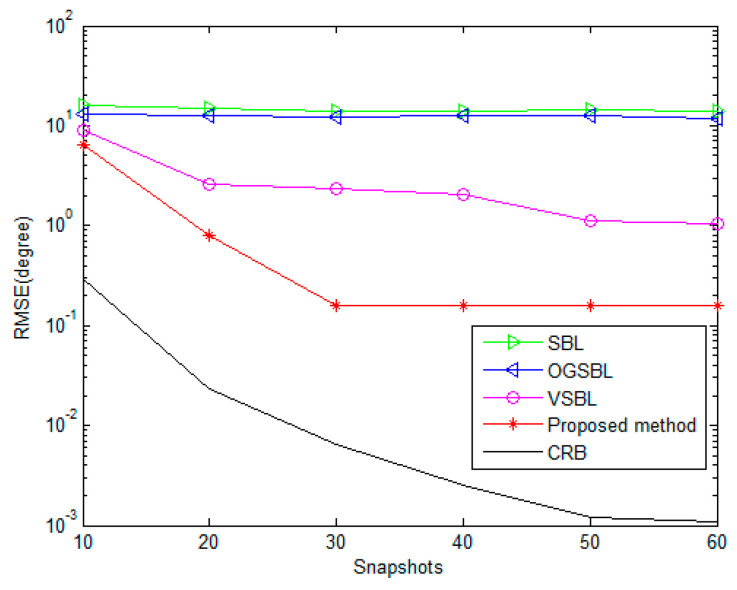
DOA RMSE versus the number of snapshots (SNR = 3 dB).

**Figure 8 sensors-24-02485-f008:**
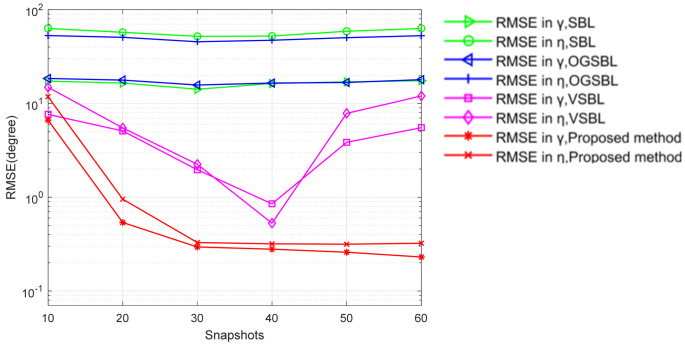
Polarization RMSE versus the number of snapshots (SNR = 3 dB).

**Figure 9 sensors-24-02485-f009:**
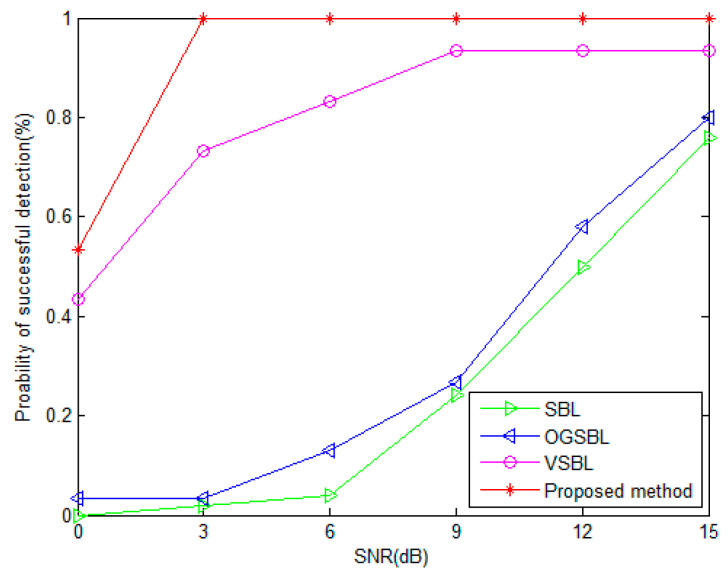
Probability of successful detection versus SNR (snapshots = 20).

**Figure 10 sensors-24-02485-f010:**
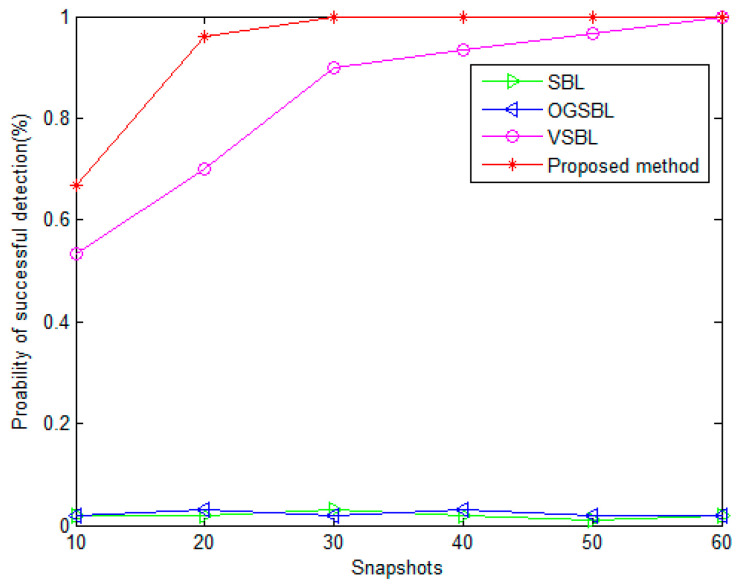
Probability of successful detection versus the number of snapshots (SNR = 3 dB).

**Figure 11 sensors-24-02485-f011:**
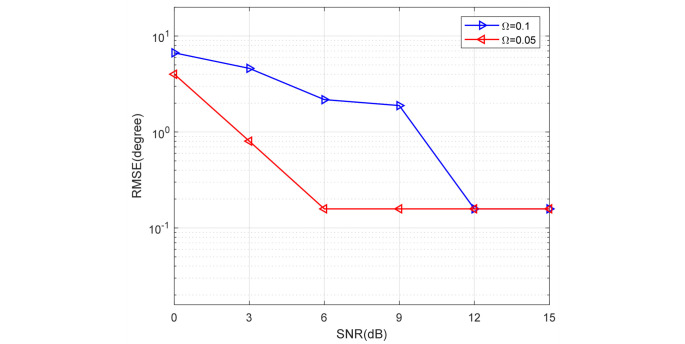
DOA RMSE versus SNR under different Ω (snapshots = 20).

**Figure 12 sensors-24-02485-f012:**
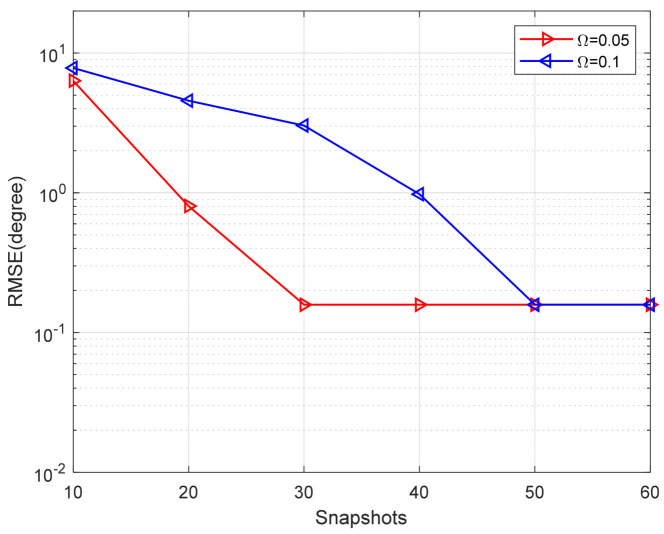
RMSE versus the number of snapshots under different Ω (SNR = 3 dB).

**Figure 13 sensors-24-02485-f013:**
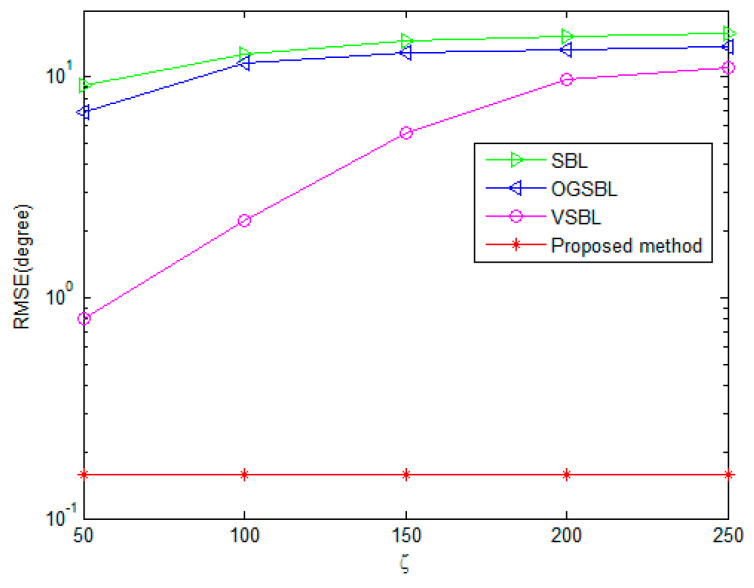
DOA RMSE versus different ς of bad data.

**Figure 14 sensors-24-02485-f014:**
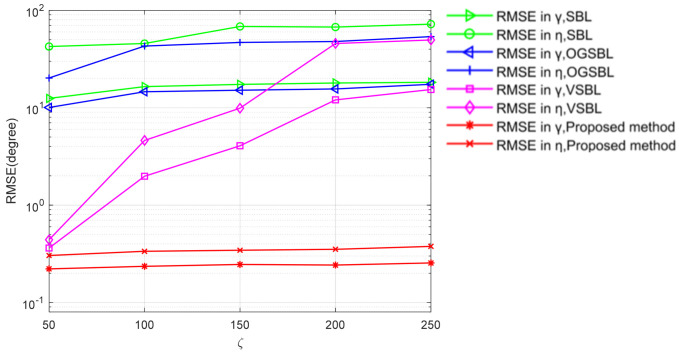
Polarization RMSE versus different ς of bad data.

**Figure 15 sensors-24-02485-f015:**
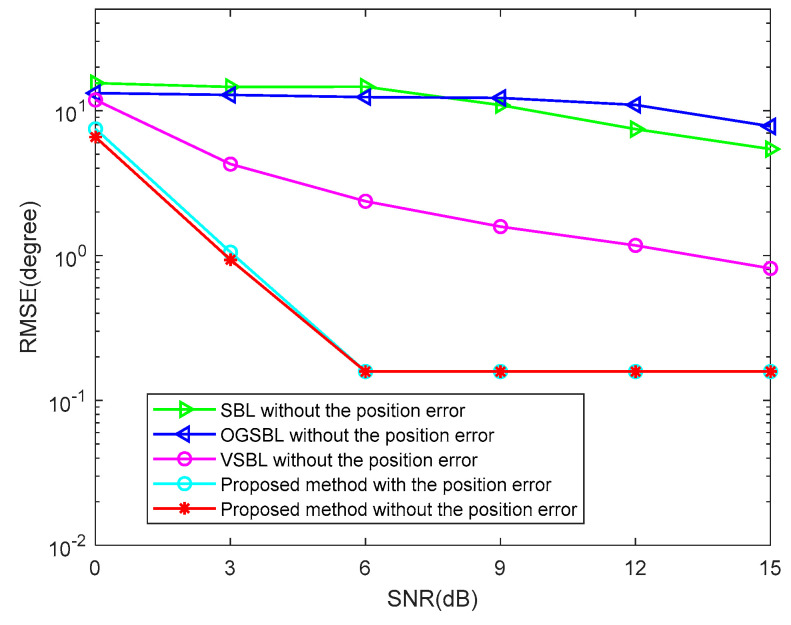
DOA RMSE versus SNR (Snapshots = 20).

**Figure 16 sensors-24-02485-f016:**
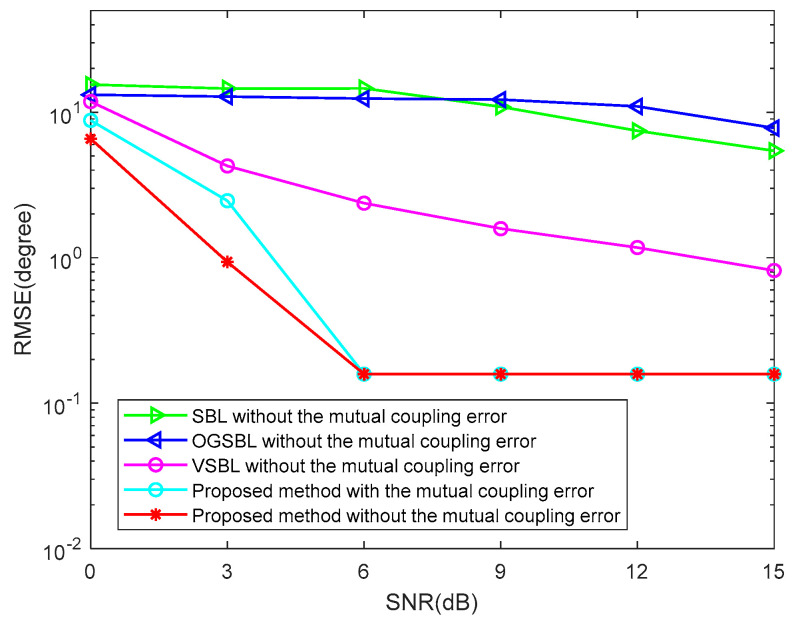
DOA RMSE versus SNR (Snapshots = 20).

## Data Availability

Data will be made available on request.

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
