# Peer review of "Robust Tensor-Based DOA and Polarization Estimation in Conformal Polarization Sensitive Array with Bad Data"

_sensors, 2024, doi:10.3390/s24082485_

Round 1
Reviewer 1 Report
Comments and Suggestions for Authors
In this paper, the authors propose a tensor variational sparse Bayesian learning method to estimate the direction of arrival (DOA) and polarization parameters jointly, based on the conformal polarization sensitive array (CPSA), considering the partially impaired sensor array. The sparse tensor-based received data model is developed for CPSA that incorporates bad data. The column vector detection method is proposed to diagnose the positions of the impaired sensors. In the cases involving partially impaired sensor arrays, a low-rank matrix completion method is employed to recover the random loss of signal information. The variational sparse Bayesian learning and minimum eigenvector methods are used sequentially to obtain the DOA and polarization parameters estimations. Finally, it is stated the Cramér-Rao bound for the proposed method with simulation results, demonstrating the effectiveness of the proposed method.
The shortcoming and missing of the paper are the following:
1. The used formulae of other authors must be accompanied by corresponding references. For example, formulae (1) and (2).
2. (Lines 214, 215): How is determined the weight value x in numerical simulations? In Algorithm 1, x = 2. How do all results change, if x ¹ 2?
3. Formula (21): aj must be replaced by 1/aj (compare, please with Line 265). All formulae in the paper should be carefully checked.
4. Formula (22): What does t mean?
5. Formula (23): What does n mean?
6. (Line 309): Reference [41] must be replaced by [40]. In the paper, there are only 40 references.
7. Formula (44): Does the second addendum right?
8. Formula (47): What does subscript Mc mean?
9. Formula (48): The operation of tr before second addendum is absent (compare, please with formula 44). It is right?
10. Formulae (49) – (51): The operations of tr before first and second addendums are absent (compare, please with formulae 44, 48). It is right?
11. (Line 357): Fig. 1 must be replaced by Fig. 3.
12. (Lines 439, 440): “Figure 12 and Figure 13 show that the DOA and polarization RMSE of the four methods deteriorate with the increase.” - Is the unfinished phrase.
13. References on the formula of Duncan-Guttman (Line 482) and on the formula of Woodbury (Line 487) are absent.
14. Conclusion should be extended with inclusion of the comparison results obtained.
Reviewer 2 Report
Comments and Suggestions for Authors
The paper titled "Title Robust Tensor-based DOA and Polarization Estimation in Conformal Polarization Sensitive Array with Bad Data" is reviewed and the work presented by the authors is very interesting and can be considered for the publication subjected to the revisions.
1. Remove the word "Title" in the title.
2. In abstract the authors had used the word "We", avoid the usage of the words I/We/Me/Ours etc.
3. The results are very well presented and the authors have to widely discuss how do robust estimation techniques help mitigate the impact of bad data or outliers in the estimation process in the result section.
4. In this work, the authors proposed the tensor variational sparse Bayesian learning for the estimation of the DOA and the column vector detection method is utilised to locate the bad data positions. It is one of the thrust research areas and the authors had presented the paper very effectively.
5. However, the authors have to address the following issues in the revised paper.
6. The authors used the word "we" in the paper many time.....avoid the usage of the words "We/I/Our/My" in the paper.
7. The authors implemented the column vector detection method in this work. But, this method is having the disadvantages of noise vulnerability, and sensitivity. The authors have to justify their selection of CVD. Compared to CVD, the other sophisticated techniques like CNNs, Histogram of oriented gradients etc. are available in the literature.
8. The simulation analysis is good, as the authors compared the performance of the proposed method with other methods in the literature.
9. The conclusion section can be revised by highlighting the major outcomes of this wok in the point by point manner, and the authors are further advised to include limitations of this work and the scope of this work in the future .
10. The references are appropriate.
Reviewer 3 Report
Comments and Suggestions for Authors
sensors-2906994
Robust Tensor-based DOA and Polarization Estimation in Conformal Polarization Sensitive Array with Bad Data
The hybrid method might be interesting. However, authors need to consider the following technical details and editorial problems to make minor revisions to the paper.
(1) Please clarify which methods are used in the manuscript to give a flowchart.
(2) Too many formulas are not well addressed as well as the pseudo code, please correct them and remove many unclear formulas.
(3) Title should be carefully checked, for example, ‘Title’, etc.
(4) Experiment works must be given to make the hybrid method presented in the manuscript ‘robust’.
Round 2
Reviewer 1 Report
Comments and Suggestions for Authors
Comment 4: I asked in respect to Formula (22): “What does t (“tau”, no “t”) mean?” Response is absent.
Comment 5: I asked in respect to Formula (23): What does n (“nu”, no “n”) mean? Response is present.
At the same time, the authors have answered on other queries and made necessary revisions. So, the response for query 4 could be made at the stage of Page Proofs and the paper may be accepted to publication.
Reviewer 3 Report
Comments and Suggestions for Authors
I am not satifing with the responses of all comments. Please carefully check all the formulas, the pseudo codes you given in the manuscript! Also, if no experimental works are given, it might not suitable for sensor journal.
Round 3
Reviewer 3 Report
Comments and Suggestions for Authors
No further comments.